# Primary Photosensitization by *Chamaecrista serpens* in Santa Inês Sheep

**DOI:** 10.3390/ani12223132

**Published:** 2022-11-13

**Authors:** Múcio F. F. Mendonça, Lais G. Caymmi, Antônio W. O. Silva, Carmo E. A. Biscarde, Ricardo D. G. Silva, Paula V. Leal, Luciano A. Pimentel, Franklin Riet-Correa, Tiago C. Peixoto

**Affiliations:** 1Postgraduate Program in Animal Science in the Tropics, Federal University of Bahia (UFBA), Av. Milton Santos 500, Salvador 40170-110, Brazil; 2Veterinary Pathology Laboratory, UFBA, Av. Milton Santos 500, Salvador 40170-110, Brazil; 3Veterinary Pathology Sector, Federal University of Recôncavo of Bahia (UFRB), Road Rui Barbosa 710, Cruz Das Almas 44380-000, Brazil

**Keywords:** primary photosensitization, necrotizing photodermatitis, intoxication in sheep, toxic plants

## Abstract

**Simple Summary:**

Photosensitization is a cutaneous disorder caused by a substance that makes the skin sensitive to solar radiation. When this substance is endogenous (phylloethrin) as a result of chlorophyll not being metabolized in the liver, by hepatic failure, photosensitization is called secondary. When the substance is of exogenous origin, photosensitization is primary. Outbreaks of photosensitization caused by the ingestion of *Chamaecrista serpens* have been described in the semi-arid region of Bahia, Northeastern Brazil, in cattle, sheep and horses. However, there are still doubts regarding the type of photosensitization. The objective of this study was to determine if the photosensitization caused by *C. serpens* is primary or secondary. The disease was reproduced in three groups of two sheep each ingesting *C. serpens* as the only food or at daily doses of 10 and 20 g per kg bodyweight. The six sheep showed lesions of photosensitization. Two control sheep were not affected. No alterations were observed in the serum activities of aspartate-amino transferase (AST) and gamma-glutamyl transferase (GGT), suggesting that liver function was adequate. No histologic lesions were found in the livers of the sheep with photosensitization, demonstrating that the disease is a primary photosensitization caused by the consumption of *C. serpens*.

**Abstract:**

This study aimed to clarify the type of photosensitization induced by *C. serpens* and to verify if the plant remains toxic after being collected and stored. Eight crossbred sheep, aged between 6 and 36 months, were divided into three groups (G1 to G3). Over 30 days, daily, G1 received an exclusive diet of *C. serpens*, and G2 and G3 received 10 g/kg/BW and 20 g/kg/BW, respectively. Two other sheep were used as controls (CG). Before administration, the plant had been harvested every 15 days. Liver biopsies and blood samples were taken from all sheep on day zero and weekly. All sheep that received the plant developed clinical signs of photosensitization, and no changes were observed in the serum activities of AST and GGT. On day 30, all sheep except Ov1 from G1 and Ov7 were euthanized and necropsied. All sheep that received the plant developed clinical signs. Macroscopic or histologic lesions were not observed in the liver. Ov 1 recovered 13 days after the end of ingestion. These results demonstrated that *C. serpens* causes primary photosensitization. It is advisable to avoid grazing on pastures invaded by the plant or to remove them from the pastures immediately after observing the first signs.

## 1. Introduction

*Chamaecrista serpens* (L) Greene, of the Fabaceae family, is a prostrate herb, native to Brazil, which can also be found in Central America and other South American countries. The genus *Chamaecrista* is one of the most complex and taxonomically diverse genera of the Brazilian flora, with 268 species, of which 223 are endemic [1,2,3,4].

The species *C. serpens* presents phytogeographic domains in the Amazon, Caatinga, Cerrado and Atlantic Forest biomes, being distributed in the northern region (Pará, Roraima, Tocantins), northeastern region (Alagoas, Bahia, Ceará, Maranhão, Paraíba, Pernambuco, Piauí, Rio Grande do Norte and Sergipe), midwestern region (Mato Grosso do Sul, Mato Grosso and Goiás) and southeastern region (Minas Gerais and São Paulo). It grows preferentially in the savannah (cerrado); however, it often occurs on the edges of roads and lawns, in sandy soils and close to the coast. It has yellow flowers and elastically dehiscent fruits, in addition to displaying varieties based on the number and size of leaflets, flowers and petals [1,2,3,4].

A previous study of this research group reproduced the photosensitizing effect of *C. serpens* in sheep and described outbreaks of this intoxication in cattle, sheep and horses raised in the semi-arid region of the state of Bahia on pastures invaded by the plant, especially during the beginning of the rainy season. *C. serpens* is popularly known as “sarninha” or “pela-égua” and presents with severe photosensitive dermatitis, which has drawn the attention of ranchers and equine breeders, due to its high incidence and, consequently, economic losses. Clinical signs observed include restlessness, irritation, pruritus and photosensitization [5].

In this recent study, the photosensitizing effect of the plant was confirmed with the introduction of sheep into pastures heavily invaded by *C. serpens* (in loco study); however, there were still doubts about the type of photosensitization caused (primary or secondary), as well as whether the plant loses its toxicity after being collected and stored, which occurs with some plants causing primary photosensitization [6,7,8]. In addition, there are still knowledge gaps regarding the evolution of this intoxication, the toxic dose, the pathogenesis and the active ingredient.

The objective of this study was to clarify the type of photosensitization induced by *C. serpens*, and to verify whether the plant remains toxic after being collected and stored. In addition, we sought to recommend forms of control and prevention of poisoning.

## 2. Materials and Methods

This study was approved by the Ethics Committee on the Use of Animals of the Federal University of Recôncavo da Bahia (protocol number 23007.013398/2012–21). All procedures followed the care recommended by the CEUA-UFRB, regarding the use of animals in research.

The experiment was carried out for 60 days between May and July 2019, at the Centro de Desenvolvimento da Pecuária (CDP) of the Federal University of Bahia (UFBA), in Oliveira dos Campinhos, district of Santo Amaro, Bahia, Brazil. Eight common Santa Inês sheep were used, one male (Ov1) and seven females (Ov2 to Ov8), with body weight (BW) between 13 and 50 kg and aged between six and 36 months. Two of these sheep were recently weaned and kept on elephant grass pasture, without the presence of toxic plants (Ov1 and Ov2 of group 1). Sheep were held indoors in four 5 × 2 m stalls (two sheep per stall). The plant was offered early in the morning, and, after the end of ingestion, the animals were transferred to a sunny area without pasture. At night, they were placed back in the stall. Before the beginning of the experiment, the animals were dewormed with 5 mg of Levamisole Hydrochloride/kg/BW, orally, and kept for one week under experimental conditions to adapt to the management. The experimental design is shown in Table 1.

The plants used in the experiments (stem, leaves, flowers and fruits) were harvested on a property in the municipality of Santaluz (11°19′27.9″ S 39°40′15.5″ W—Figure 1a), where spontaneous outbreaks of photosensitization caused by *C. serpens* were recently described in cattle, sheep and horses [5]. Exsiccate specimens of the plant used in the experiments were identified in the Herbarium of the Universidade Estadual de Feira de Santana (HUEFS) (n° 240995) as *Chamaecrista serpens* (L) Greene var. *serpens* (Figure 1b). The plant was collected every 15 days, stored in strain bags and kept in a cold chamber (3–5 °C), as previously described [8]. Daily, the dose of the plant administered to the animals of the experimental groups G1, G2 and G3 was weighed, 12 h before the supply, kept at room temperature and administered in individual troughs for spontaneous consumption at 7 a.m.

The eight experimental sheep were divided into four groups of two sheep each. G1 received *C. serpens* as the only food for 30 days. Groups G2 and G3 received, daily, 10 and 20 g/kg, respectively, of *C. serpens* mixed with the roughage. Sheep from G4 were used as controls. All animals, except those in G1, received a maintenance diet consisting of elephant grass (*Pennisetum purpureum*) and corn silage, in addition to commercial sheep feed (1% of BW). In addition, mineral salt suitable for the species and water ad libitum were offered to all animals. Two of the sheep (Ov1 and Ov2, both of G1) were recently weaned and the entire herd was maintained before the experiment in a pasture free of *Brachiaria* spp. and other hepatotoxic plants.

The sheep were clinically examined daily, with detailed skin inspection. Weekly, on days zero (D0), seven (D7), fourteen (D14), twenty-one (D21) and twenty-eight (D28), they were individually weighed for eventual adjustment of the daily dose ingested. On these occasions, they were subjected to blood collection by jugular vein puncture, using a vacuum system (Vacutainer^®^, Becton Dickinson Surgery Industries Ltd., Curitiba, Brazil), in 5 mL tubes with sodium ethylenediaminetetraacetate (EDTA) anticoagulant in 10% aqueous solution, to perform the hemogram according to the methodology described by Jain [9], and in 10 mL tubes containing coagulation activator and separator gel. Ov1 from G1 and Ov7 from CG were subjected to additional blood collections, on D45 and D60, during the assessment of recovery from photosensitive lesions.

Serum samples were obtained after clot retraction, with centrifugation at 1200× *g* for 10 min and, later, placed in polyethylene microtubes (Eppendorf^®^, Hamburg, Germany) and sent for measurement of serum activities of aspartate-amino transferase (AST) and gamma-glutamyl transferase (GGT), by the kinetic enzyme method, according to the methods described for the PKL 125^®^ automatic biochemical analyzer (MH Equipment and Material for Laboratories Ltd., São Paulo, Brazil). The individual values on D0 and those recommended by Kaneko et al. [10] were used as a reference.

All sheep underwent histologic evaluation of the liver using ultrasound-guided biopsies with a number 14 catheter on D0, using a previously reported technique [11]. The animals were kept stationary, and the catheter was inserted into the 11th intercostal space, after local trichotomy (10 cm^2^), surgical antisepsis and local blockade, using an anesthetic button, with 5 mL of 2% lidocaine hydrochloride (Dorfin^®^, Ceva Animal Hearth, Paulinia, Brazil), in the intercostal muscle. At least three liver samples were collected from each animal.

On D30, the supply of the toxic plant to sheep in groups G1–G3 was stopped. On this occasion, all animals, except Ov1 and Ov7, were subjected to euthanasia and necropsy. During the necropsy, fragments of the skin from different regions (with and without lesions), organs from the abdominal, thoracic and pelvic cavities and the brain were collected. This material was fixed in 10% formalin, processed by routine histological techniques, stained with hematoxylin and eosin (HE) and evaluated under an optical microscope.

Additionally, after cessation of plant administration, Ov1 from G1 and Ov7 from CG were relocated to a shaded paddock and followed for another 30 days to assess the recovery of skin lesions of Ov1. During this phase, these animals received elephant grass (*Pennisetum purpureum*) and commercial sheep feed (1% of BW), in addition to mineral salt and water ad libitum.

## 3. Results

Sheep from G1–G3 showed good avidity for all parts of the plant (leaf, flower, fruit and stem), ingesting them spontaneously; however, from D16 onwards, Ov6 showed a capricious appetite and started to reject the plant (hyporexia), and its ingestion was minimally forced. All sheep that received the plant developed dermatitis of varying intensity, compatible with photosensitization. The evolution of the intoxication, with an emphasis on the intensity of clinical signs, is described in Table 2.

From D3 onwards, Ov2, which had a white coat and consumed exclusively *C. serpens*, showed marked clinical signs, characterized by restlessness, itching and intolerance to sunlight. Ov1, from the same experimental group (G1), however with black fur, showed similar clinical signs after D5, but of mild intensity. On the same day, the G3 animals presented only mild (Ov6—black and white coat) to moderate (Ov5—red coat) itching. This sign was only observed in Ov3 and Ov4 (G2), both with black and white fur, from D9 and D23, respectively, with moderate intensity.

The restlessness observed in the animals of G1 was similar, but with more intensity. In general, G1 sheep showed behaviors such as jumping, running, lying down and standing up frequently, as well as kicking the ground and shaking their heads. On days of higher solar incidence, clinical signs were exacerbated. This concern was more discreet throughout the experiment in G2 and G3, but on D7 and D23 (intense sunny day), Ov5 and Ov4, and Ov5 on D7 and Ov4 on D23 (intense sunny days), also showed this clinical sign.

Along with restlessness, itching was widespread and pruritus was widespread and accentuated in Ov2 and Ov5. In Ov1, Ov3, Ov4 and Ov 6, this clinical sign occurred in a mild to moderate way, being generalized in Ov1. In the others, the lesions were mainly around the trichotomy, axilla, groin and depigmented regions of the skin. However, all these clinical signs were minimized on rainy days, as the animals remained in the pens. Other acute clinical signs occurred exclusively in Ov2 from D5 onwards, which presented edema and facial erythema, especially in the nostrils and ears (Figure 2a). Simultaneously, the sheep exhibited an antalgic position characterized by “lowering the hip” and walking in circles (Figure 2b). Restlessness, moaning, pain on palpation, congested mucous membranes, tachycardia, tachypnea and hyperthermia were also observed. Ov4 also showed severe signs, but at different times, showing intolerance to sunlight and photosensitization (Figure 3d) on D16 and D23, respectively.

Overall, all sheep that received the plant exhibited erythema and cutaneous edema prior to the characteristic photosensitization lesions. There was a marked difference in the dimension and intensity of the skin lesions between each sheep. Ov2 exhibited more severe generalized skin lesions on the face, ears and mucocutaneous junctions (Figure 3b). In addition, on D7, it presented edema in the pectoral region. On the other hand, Ov1 did not exhibit serious skin lesions, which can be explained by its black coat; however, on D24, it was noted that the hairs were brittle, with a fall in the dorsal region and discreet alopecia in the chamfer region, in addition to slightly upturned ears and edematous blepharitis (Figure 3a). Ov3 presented discrete edema and erythema in the axilla and groin region on D9, with depigmented areas, which evolved into dermatitis on D23 (Figure 3c). Similar signs were observed in Ov6, mainly in the chest region, where shaving was performed for liver biopsy. The sheep in the CG did not show clinical signs of intoxication.

After ceasing the administration of the plant to Ov1 of G1 on D30 and relocating it to a shaded paddock, there was a gradual recovery of the skin lesion, with full recovery after 13 days.

The assessments of serum AST and GGT activities performed on D0 of sheep from G1, G2 and G3 and sheep from the CG showed no differences when compared to those performed throughout the experiment (D7, D14, D21 and D28) in sheep poisoned by the plant. Thus, no signs of liver damage were observed in all sheep with photosensitization lesions caused by the consumption of *C. serpens* (Table 3). Furthermore, after stopping the supply of the plant for Ov1 in G1, the values of AST and GGT at D28, D45 and D60 remained similar to those observed during the experiment (D7, D14, D21 and D28). In all these liver function assessments, the results were within the reference values and similar to those of the CG, as shown in Table 4.

The necropsy performed on five sheep intoxicated by *C. serpens* (Ov2 to Ov6) allowed a better assessment of the skin lesions observed during the clinical examination (necrotizing ulcerative dermatitis with hyperemia, alopecia, scaling and areas of scarring and dermal thickening). These lesions occurred mainly in the thorax in the area near the trichotomy, axilla, groin and depigmented areas (pelvis, back, udder and tail crease), where their intensity varied from mild to severe. Additionally, Ov2 presented areas of crusted alopecia lesions on the head, especially on the snout, nasal plane, ears and upper and lower left eyelids, which completely covered the eyeball due to marked eyelid edema; there was also marked corneal opacity and focally extensive ulcer and seromucous secretion covering the eyeball (Figure 4). The sheep in the CG showed no skin lesions.

Significant macroscopic changes were not observed in the liver or other organs of the animals intoxicated by *C. serpens*, similarly to what was observed in the control group.

The histopathological evaluation of the skin of the five sheep poisoned by *C. serpens* (Ov2 to Ov6) that were necropsied revealed characteristic changes in photosensitization (Figure 5). In general, superficial orthokeratotic hyperkeratosis, marked hypergranulosis, pseudoepitheliomatous hyperplasia (acanthosis), multifocal vacuolization of keratinocytes and extensive areas of epidermal necrosis with ulceration and serocellular crusts were observed. In the superficial and deep dermis, there was mild edema, marked angiogenesis, intense diffuse proliferation of fibroblasts and disorganized collagen fibers, as well as mild to moderate multifocal mononuclear periadenitis and sweat gland ectasia.

No histological lesions were observed in the biopsies or in the postmortem examination in the livers and other organs of sheep that showed photosensitization (G1 to G3) or in the control group.

## 4. Discussion

Recently, the toxicity of *C. serpens* was described in ruminants and horses in the semi-arid region of Bahia and its photosensitizing action was confirmed in sheep [5]. In these initial studies, the intoxication was produced in adult sheep in a pasture invaded by *C. serpens*. In this experiment, only one necropsy was performed, and the animal showed liver lesions suggestive of hepatogenous photosensitization. However, in this same study [5], the authors drew attention to some inconsistencies: the appearance of clinical signs was very fast, lethality in natural cases was rare, and there was rapid recovery after removal of the animals from the pastures invaded by the plant.

In the present complementary study, to clarify the type of photosensitization caused by the plant, new experiments were carried out, with the administration ad libitum or at doses of 10 and 20 g of the plant per kg body weight during 30 days to six sheep, including newly weaned sheep (without previous consumption of any plants or potentially hepatotoxic substances). In this research, serial biochemical evaluations (liver function) did not show any liver damage, which, added to the absence of macroscopic and histological liver changes, ruled out the initial suspicion of hepatogenous photosensitization. Similar findings are described in cases of primary photosensitization caused by *Froelichia humboldtiana* [8,12,13], *Malachra fasciata* [14], *Ammi majus* [15] and *Heterophyllaea pustulata* [16].

In this way, it becomes clear that *C. serpens* intoxication produces a picture of primary photosensitization. It should be noted that a similar event occurred when natural outbreaks of photosensitization caused by *Froelichia humboldtiana* were reported for the first time in ruminants and equines in the semi-arid region of Rio Grande do Norte [17]. On this occasion, there was a correct, unprecedented finding of the photosensitizing effect of the plant, but it was misinterpreted as secondary photosensitization. These two situations reinforce the importance of a more comprehensive and representative evaluation (larger number of animals and controlled experiments) of clinical–epidemiological, laboratory and anatomopathological findings to elucidate the pathogenesis of new toxic plants causing photosensitization.

In the pioneering study [5], the probable hepatoxic action of *C. serpens* was based, equivocally, on the increase in the activity of the GGT enzyme, as well as the observation of degenerative–necrotic alterations in liver biopsies and in one necropsied animal. After the present experiments, we verified the non-repetition of these findings, which, apparently, could be associated with the consumption of another plant or hepatotoxic agent previously in the pastures. The supply of the plant in the trough in a controlled manner and the use of some recently weaned sheep (G1) excluded any interference of this nature.

Furthermore, the occurrence of artifacts from liver biopsies mimicking degenerative–necrotic changes in the tiny samples collected cannot be ruled out [5]. It is noteworthy that, in the present study, no significant histological differences were observed in the livers of sheep that presented photosensitization (G1 to G3) and those of the CG, nor between the liver samples of the treated sheep, collected before or during the experiment or at necropsy.

The clinical evolution of *C. serpens* intoxication observed in the sheep of the present study was similar to that described previously, in naturally intoxicated cattle, sheep and horses, as well as in sheep experimentally kept on pastures invaded by *C. serpens* [5]. In the previous study, the first clinical signs occurred seven days after the introduction of sheep in pastures invaded by the plant [5]. In sheep that received an exclusive diet of *C. serpens* (G1), the first clinical signs occurred from the third day of consumption, which shows the great toxic potential of the plant.

Recently [14], it was demonstrated that the exclusive consumption of *Malachra fasciata* also caused signs of photosensitization in sheep, seven days after the beginning of ingestion of the plant. The photosensitizing effect of *F. humboldtiana* was reproduced in sheep and cattle, respectively, 10 and 3 days after the animals were introduced to pastures invaded by the plant [8,9,10,11,12,13].

To date, the toxic compound of *C. serpens* responsible for its photosensitizing effect is unknown. However, in phytochemical studies, anthraquinone compounds have already been isolated from plants of the genus *Chamaecrista* [18]. Anthraquinones present in the fruits and seeds of *Chamaecrista fasciculata* and *Chamaecrista nictitans* can cause irritation of the digestive tract when ingested in large amounts by cattle and sheep, although these disorders are often rapid and self-limiting [19]. It is important to remember that anthraquinones have also been isolated from *Heterophyllaea pustulata* and *Heterophyllaea lycioides,* plants that are known to cause primary photosensitization and eye lesions in sheep in Argentina, where the disease is popularly known as “cegadeira” (causing blindness) [16,20]. In the present study, no digestive changes were observed.

On the other hand, the toxic compounds of some plants that cause primary photosensitization in Brazil are already known. The photosensitizing effect of *Ammi majus*, which induces cutaneous and oculopalpebral lesions [15], is attributed to furocoumarins [6]. As for *Fagopyrum esculentum* and *Hypericum perforatum*, despite the absence of natural cases of intoxication in the country [21], their toxic effects are caused by naphthodianthrones derivatives, called fagopyrin and hypericin, respectively [6]. There is evidence that the toxicity of *F. humdoldtiana* is also related to naphthodianthrones or similar substances, since, when removed from the soil, it loses its toxicity and causes no eye lesions [8].

In the present study, the eye and eyelid lesions observed in Ov2 were attributed to self-inflicted trauma because of severe pruritus. Thus, it seems that in *C. serpens* intoxication, there are no primary eye lesions (corneal edema, keratoconjunctivitis and blindness), as occurs in cases of primary photosensitization caused by plants that have furocoumarins or anthraquinones as a toxic component [16,20,22]. Additionally, in *C. serpens* poisoning, only skin lesions were observed, as in the case of photosensitization attributed to naphthodianthrones [22].

On the other hand, it is interesting to emphasize that, in the present study, *C. serpens* did not lose its toxicity after being collected and stored in a cold chamber for up to 15 days. The loss of the ability to cause intoxication after harvesting and haying has already been demonstrated in *Hypericum perforatum*, which contains naphthodianthrones as a toxic component [7]. The toxicity of *F. humboldtiana* is quickly lost after collection, which justifies the failure of experimental reproduction in sheep, horses and donkeys that received the plant for at least 30 days, after being collected and kept in a cold chamber for periods of 1 to 4 days [8]. Thus, studies that aim to identify the toxic components of *C. serpens* and *F. humboldtiana* are necessary.

In the five necropsied sheep, the most important alterations were cutaneous, with macroscopic lesions characteristic of photosensitization being observed, especially in areas of the skin most exposed to sunlight and lacking hair protection or depigmented. The histopathological findings of the skin observed in this study were similar to those reported in intoxications by *F. humboldtiana* and *M. fasciata*, which can also cause severe diffuse ulcerative and suppurative dermatitis (photosensitization) [8,14].

The diagnosis of intoxication by *C. serpens* should be based on the history and presence of the plant in pastures, associated with clinicopathological findings. However, the differential diagnosis must be performed with primary and secondary photosensitization caused by other photosensitizing plants.

As a treatment and prophylaxis measure for poisoning, it is recommended to remove the animals from pastures invaded by *C. serpens* and/or transfer them to shaded paddocks free from photosensitizing plants [5]. Furthermore, dark-haired animals are more resistant to intoxication and have, generally, less severe skin lesions; therefore, they can be used as an alternative to help in plant control in invaded pastures.

Despite the use of corticosteroids being indicated to relieve itching, restlessness and reduction of edema, as well as daily cleaning and dressing of skin lesions, in addition to the topical use of repellent-healing ointment [23], in the present study, Ov1 fully recovered, 13 days after ceasing the consumption of the plant, only allowing the sheep access to the shade. However, in severe cases, treatment of skin lesions is recommended for secondary infections and myiasis.

We emphasize the need to include *C. serpens* poisoning in the list of differential diagnoses for cases of photosensitive dermatitis and the importance of research with toxic plants in Brazil, since, every year, new poisonings by plants of interest for livestock are discovered, currently totaling 148 species in Brazil, of which 64 are in the northeastern region [5,14,21,24].

## 5. Conclusions

The absence of biochemical alterations (GGT and AST) and significant histological hepatic lesions in sheep with photosensitization lesions rules out a hepatotoxic action of *C. serpens*. However, macro- and microscopic skin lesions were characteristic of photosensitization. Thus, it is clear that *C. serpens* causes primary photosensitization and that the ingestion of daily doses from 10 g/kg is capable of causing clinical signs of intoxication and photosensitization.

Immediate removal of affected animals from pastures and keeping them in the shade until the lesions disappear are indicated for the control of *C. serpens* intoxication. As a prophylactic measure, it is advisable not to place animals, especially those with areas of depigmented skin, in pastures invaded by *C. serpens*, especially during the plant’s vegetative period, which occurs after the onset of rains. Additional studies should be carried out to determine the toxic compound of the plant.

## Figures and Tables

**Figure 1 animals-12-03132-f001:**
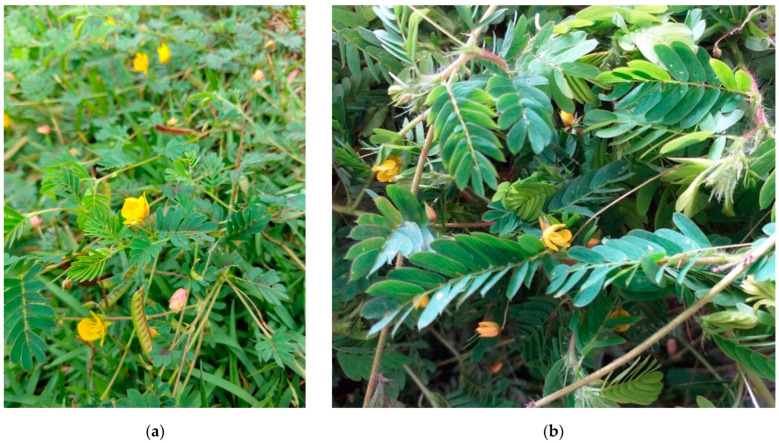
Experimental poisoning by *Chamaecrista serpens* in sheep. Samples of *Chamaecrista serpens* in flowering and fruiting. (**a**) *C. serpens* in native pasture. Santaluz, BA, property where outbreaks of natural intoxication occurred. (**b**) Fresh plant used in the experiments.

**Figure 2 animals-12-03132-f002:**
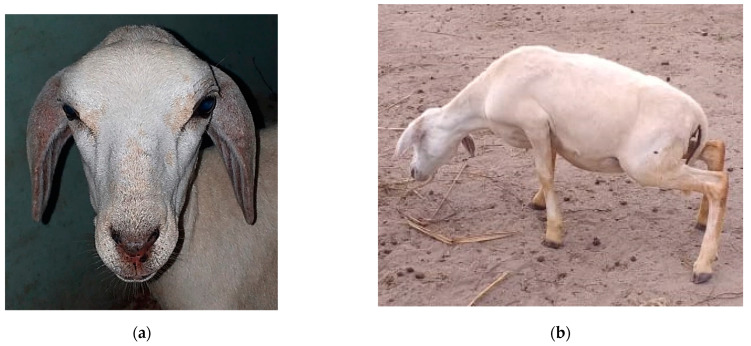
Experimental poisoning by *Chamaecrista serpens* in sheep. Ov2 from G1. Day 5. (**a**) Facial eczema. (**b**) Postural change (sign of pain) and walking in circles.

**Figure 3 animals-12-03132-f003:**
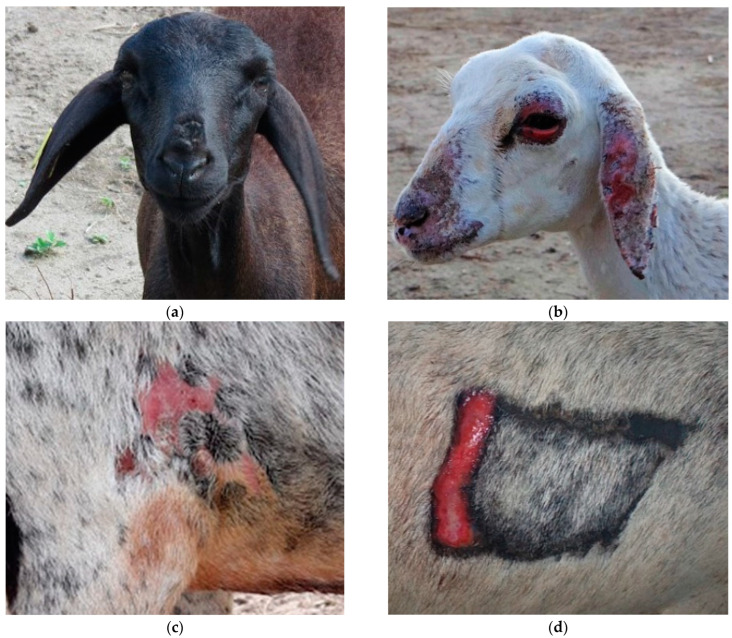
Experimental poisoning by *Chamaecrista serpens* in sheep. (**a**) Ov1 with slightly upturned ear tips, chamfer alopecia and edematous blepharitis. Day 23. (**b**) Ov2 on day 10, severe alopecic and crusted skin lesions on face, ears and mucucutaneous junction. (**c**) Ov3 on day 23, photosensitization lesions in depigmented areas of the axillary region. (**d**) Ov4 on day 23, photosensitization in the trichotomy region.

**Figure 4 animals-12-03132-f004:**
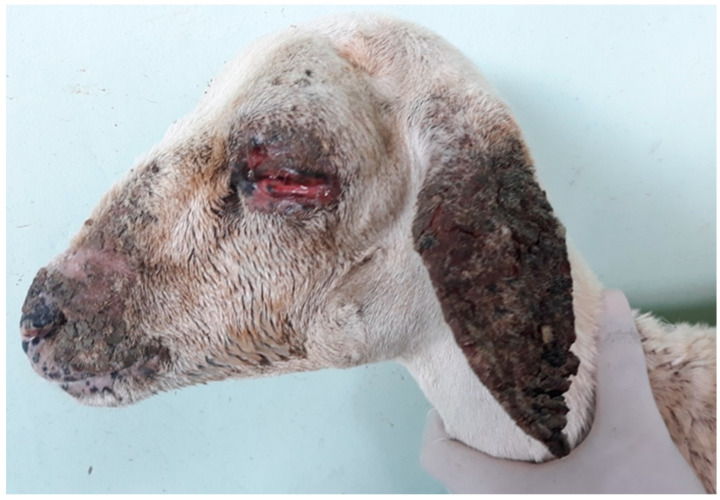
Experimental poisoning by *Chamaecrista serpens* in sheep. Marked photosensitization in Ov2. Note crusted alopecia lesions on the snout, nasal plane and ear, associated with marked eyelid edema.

**Figure 5 animals-12-03132-f005:**
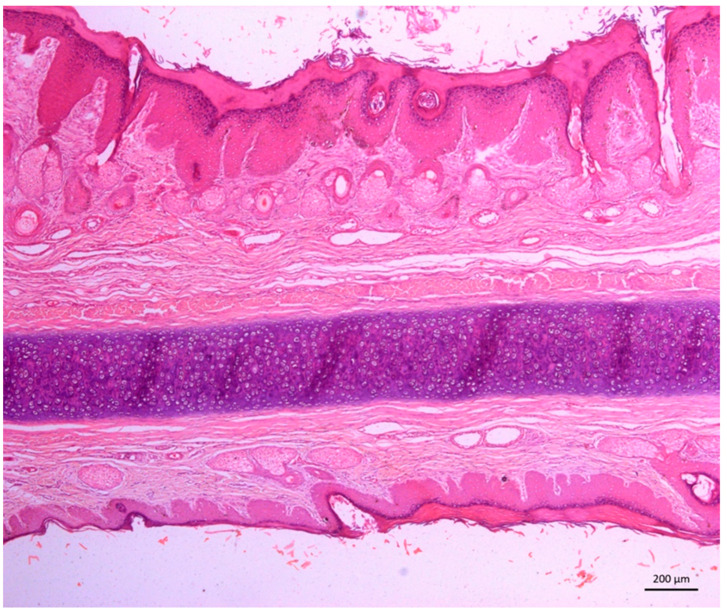
Experimental poisoning by *Chamaecrista serpens* in sheep. Photomicrography. Note marked hyperkeratosis and acanthosis associated with mild proliferation of fibroblasts and collagen fibers in the superficial dermis of the dorsal surface of the ear skin. Ov2. HE. Obj. 5x.

**Table 1 animals-12-03132-t001:** Experimental poisoning by *Chamaecrista serpens* in sheep. Experimental design and outcome.

Group	ID	Fur	Starting Weight (kg)	Final Weight (kg)	Daily Dose	TV (kg)	Onset of Clinical Signs	Degree of Photosensitization	Outcome
G1	Ov1	Black	15	14	Exclusive	45	D5	Mild	Recovery
	Ov2	white	15	13		42	D3	Severe	Euthanasia
G2	Ov3	Black and white	40	37	10 g/kg/BW	11.5	D9	Moderate	Euthanasia
	Ov4	Black and white	42	40		12.3	D14	Severe	Euthanasia
G3	Ov5	Red	36	34	20 g/kg/BW	21	D5	Mild	Euthanasia
	Ov6	Black and white	36	34		21	D5	Moderate	Euthanasia
CG	Ov7	Light red	49	50	--	--	--	--	Not intoxicated
	Ov8	Black and white	38	38	--	--	--	--	Euthanasia

Legend: ID = sheep identification; TV = total volume ingested.

**Table 2 animals-12-03132-t002:** Experimental poisoning by *Chamaecrista serpens* in sheep. Main clinical signs, evolution and intensity of intoxication.

Clinical Signs	GROUPS							
	G1		G2		G3		CG	
	Ov1	Ov2	Ov3	Ov4	Ov5	Ov6	Ov7	Ov8
Restlessness *	D5 ++(+)	D3 +++	D27 +	D23 ++	D7 ++	D15 +	(-)	(-)
Pain signs	D6 +	D5 +++ ^α^	(-)	(-)	(-)	(-)	(-)	(-)
Postural change	D6 +	D5 +++ ^β^	(-)	(-)	(-)	(-)	(-)	(-)
Intolerance to sunlight **	D6 +	D3 +++	(-)	D16 +++	D16 +	(-)	(-)	(-)
Itching	D5 +(+)	D3 +++	D9 (+)	D23 +(+)	D5 ++(+)	D5 +(+)	(-)	(-)
Apathy/lethargy	D6 +	D6 +++	(-)	D20 (+)	D18 (+)	D16 ++	(-)	(-)
Hyperemic mucous membranes	D5 ++	D5 +++	D27 (+)	D23 +	(-)	D15 (+)	(-)	(-)
Facial eczema ***	(-)	D5 +++	(-)	(-)	(-)	(-)	(-)	(-)
Edema and erythema	D16 (+) ^1^	D7 ++ ^2^	D9 ++ ^3^	D14 (+) ^4^	D10 (+) ^4^	D10 +(+) ^3,4^	(-)	(-)
Photosensitization	D23 (+) ^a^	D5 +++ ^b^	D23 ++ ^c^	D23 +++ ^d^	D10 (+) ^d^	D14 ++ ^c^	(-)	(-)
Hyporexia	(-)	(-)	(-)	(-)	(-)	D16 +++	(-)	(-)
Slimming	D30 +	D15 ++(+)	D30 ++	D30 ++	D30 ++(+)	D15 ++(+)	(-)	(-)

Legend: G1 = exclusive diet, G2 = 10 g/kg of *C. serpens*, G3 = 20 g/kg of *C. serpens*, CG = control group; DE = day of experiment; intensity of clinical signs: (absent (-), mild (+), mild +, mild to moderate +(+), moderate ++, moderate to severe ++(+) or severe +++; * = kicks the floor, lies down and gets up, jumps, runs, shakes head; ** = attempt to avoid exposure to sunlight; *** = edema and erythema on the face, nostrils and ears; ^α^ = tremors, groans and bruxism; ^β^ = weakness; ^1^ = ears and eyelids, ^2^ = edema of the chest, ^3^ = armpit, groin and depigmented areas; ^4^ = side; ^a^ = brittle hair on the back and slight lesion in the nostrils, ^b^ = generalized, ^c^ = axilla, groin, side and croup (depigmented areas), ^d^ = side (area of trichotomy).

**Table 3 animals-12-03132-t003:** Experimental poisoning by *Chamaecrista serpens* in sheep. Serum aspartate aminotransferase (AST) and gamma-glutamyl transferase (GGT) activities of six sheep experimentally intoxicated by *Chamaecrista serpens*.

Group	ID	Diet	Enzymes (U/L)	Harvest Day					Reference Values *
				D0	D7	D14	D21	D28	
G1	Ov1	Exclusive	AST	113.3	61.7	80	56.7	77.92	60–280 U/L
			GGT	38.76	46.39	42.37	37.55	41.26	20–52 U/L
	Ov2		AST	91.1	55.7	84	65.8	74.15	60–280 U/L
			GGT	51.61	34.54	52.61	46.59	46.34	20–52 U/L
G2	Ov3	10 g/kg/BW	AST	60.7	54.7	69.8	78.9	67.8	60–280 U/L
			GGT	56.43 ^a^	55.63 ^a^	64.26 ^a^	57.84 ^a^	52.61 ^a^	20–52 U/L
	Ov4		AST	74.9	78.9	81	67.49	72.9	60–280 U/L
			GGT	64.66 ^a^	67.88 ^a^	63.06 ^a^	105 ^a^	64.66 ^a^	20–52 U/L
G3	Ov5	20 g/kg/BW	AST	89.1	87	88	74.9	80	60–280 U/L
			GGT	48.23	44.38	45.75	37.75	41.57	20–52 U/L
	Ov6		AST	57.7	78.9	56.7	45.5	48.6	60–280 U/L
			GGT	47.59	47.8	46.59	40.36	41.37	20–52 U/L
CG	Ov7	NRP	AST	74.5	93.1	65.3	90	87.3	60–280 U/L
			GGT	55.41 ^a^	40.2	50.1	52.2 ^a^	53 ^a^	20–52 U/L
	Ov8		AST	61.7	92.2	90.1	89.4	88	60–280 U/L
			GGT	35.7	50.2	46.4	49.1	47	20–52 U/L

* Kaneko et al. (2008), pp. 882–884; NRP = did not receive the plant; ^a^ values above the reference.

**Table 4 animals-12-03132-t004:** Experimental poisoning by *Chamaecrista serpens* in sheep. Serum activity values of gamma-glutamyl transferase (GGT), aspartate aminotransferase (AST), total plasma protein (TPP) and fibrinogen (FIB) of Ov1 and Ov7, in the recovery phase of intoxication by *Chamaecrista serpens*.

ID	Enzymes	Harvest Day			Reference Values *
		D30	D45	D60	
Ov1	AST (U/L)	80	56.7	130.5	60–280
	GGT (U/L)	42.41	42.37	37.55	20–52
	TPP (g/dL)	5.6	6	6.2	6–7.5
	FIB	400	200	200	100–500
Ov7	AST (U/L)	65.3	70.1	64.8	60–280
	GGT (U/L)	50.1	45.4	48.2	20–52
	TPP (g/dL)	6	6.2	6.5	6–7.5
	FIB	100	200	100	100–500

* Kaneko et al. (2008), pp. 882–886.

## Data Availability

Not applicable.

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
