# Peer review of "Primary Photosensitization by Chamaecrista serpens in Santa Inês Sheep"

_animals, 2022, doi:10.3390/ani12223132_

Round 1
Reviewer 1 Report
The article is suitable for publication,has relevance and belongs to the scope of the journal. It is suggested to the autors to detail the reason for not considering facial eczema as a sign of photosensitization
Author Response
The denomination facial eczema is correlated to the disease caused by the ingestion caused by Pithomyces chartarum. This fungus, saprophytic on certain grasses and forage legumes. It is already confirmed that in Brazil there is no incidence of this agent in sufficient proportions to cause the lesion. Therefore, this term is not used very often.
Reviewer 2 Report
The presented paper reports effect of experimental intoxication of sheep by ingestion of C. serpens in order to explain the type of photosensitization caused by this plant and to verify if it remains toxic after harvesting and storage. The manuscript seems to be clearly written, however experiment design is inappropriate or, at least, description of the methodology is insufficient and contains mistakes.
General remarks:
- Lines 87-88: “Eight crossbred Santa Inêsand Dorper sheep were used, one male (Ov1) and seven females (Ov2 to Ov8), with live weight (LW) between 10 and 40 kg and age between six and 36 months.” - The study was conducted on a very small number of sheep, in addition animals were differentiated in terms of age, sex, physiological status, coat color, etc. All of these factors can influence light sensitivity. It is well known that young animals and light-colored individuals are more susceptible to skin damage under the influence of sunlight. Ov2 that showed the most severe clinical symptoms was “newly weaned” and white. Please argue that the experimental groups were representative.
- Lines 91-92: “During the day the sheep were kept in a sunny area without pasture and at night” - what does it mean? Part of the sentence is probably missing here. Additionally, please describe in more detail the conditions in which the animals were kept. Were the animals from different groups kept in exactly the same conditions and in the same pen? Were the animals able to hide in the shade? Were the animals trimmed prior to the start of the experiment? If so, are all of them? At the same time? In the same way?
- Line 92: “The animals were dewormed before the beginning of the experiment” - Please provide detailed information on deworming (name of the drug, dose, term of deworming), bacause antiparasitic agents can act photosensitizing.
- Line 101: “The plant was collected every 15 days, stored in strain bags, and kept in a cold chamber (3-5°C).” - Are these conditions suitable for storing plants for animal nutrition? Were the feed rations for the control animals also stored in this way? Is there a risk of microbial growth on plants during cold storage, which could have an adverse effect on animal health?
- Lines 117-118: “Four of the sheep were recently weaned and all the herd were maintained previous to the experiment in a pasture free of Brachiaria spp and other hepatotoxic plants.” - Please indicate exactly which animals were just after weaning. The body weight of 6 out of 8 individuals indicates that they were not individuals just after weaning. Why were the youngest animals taken into group G1? Please explain it.
- Lines 130-133: “Serum samples were obtained after clot retraction, with centrifugation at 1200X g for 10 minutes and, later, placed in polyethylene microtubes (Ep-pendorf®) and sent for measurement of serum activities of aspartate-amino transferase (AST) and gamma-glutamyl transferase (GGT)”. - Why did you not perform a biochemical analysis of total bilirubin, direct bilirubin, indirect bilirubin, while your pilot study (referred to in reference 5) showed significant changes in these parameters in sheep experimentally intoxicated with C. serpens? Please explain it.
- Lines 144-148: "During the necropsy, fragments of the skin from different regions (with and without lesions), organs from the abdominal, thoracic, and pelvic cavities, and the brain were collected. This material was fixed in 10% formalin, processed by routine histological techniques, stained with hematoxylin and eosin (HE) and evaluated under an optical microscope.” - What was the purpose of taking samples of the brain and organs from the abdominal, thoracic, and pelvic cavities? What kind of changes were expected to be found there? Please explain it.
- Line 231: Please provide more information on reference standards for the parameters described; do they take into account age, gender, etc? What is more, I have not found in the quoted paper such a range of standards for sheep in relation to GGT...
- Line 287-288: “including newly weaned sheep (without previous consumption of any plants or potentially hepatotoxic substances)” - Is it possible for lambs around 180 days (6 months) old to eat exclusively mother's milk? Have the “newly weaned lambs” been in any contact with their mothers? Is it possible that the changes observed in behavior of sheep from G1 group were related to the weaning effect? Please explain it.
Minor remarks:
1. Lines 31-32: “During 30 days, G1 received an exclusive diet of C. serpens, and G2 and G3 received 31 10g/kg/d and 20g/kg/d, respectively” - It should be mentioned that C. serpens was added on kg of body weight; it is not clear in this statement.
2. Line 33: “for x-xx days” - what does it mean?
3. In lines 88-89 is written “sheep […] with live weight (LW) between 10 and 40 kg”, while in Table 1 where is noticed that the lowest weight of sheep was 13 kg (final weight of Ov2) and the highest weight was 50 kg (final weight of Ov7). Mature weights of the Santa Ines ewes fall between 40 and 50 kg. The Dorper is the second largest breed in South Africa and adult ewes usually weigh 60-80 kg. What is typical weight of adult crossbred Santa Inês x Dorper sheep? Please provide reliable information about the weight of sheep.
4. Lines 149-150: „Additionally, after cessation of plant administration, Ov1 from G1 and Ov7 from GC were relocated to a shaded paddock and followed for another 30 days to assess the recovery of skin lesions of Ov7.” - Are there any skin lesions in Ov7? Why?
5. Lines 280-282: “However, in this same study, [14] the authors drew attention to some inconsistencies: the appearance of clinical signs was very fast, lethality in natural cases was rare and there was rapid recovery after removal of the animals from the pastures invaded by the plant.” - It is not the same study as cited in previous lines. It focused on other plant Malachra fasciata.
In lines 276-280 is written “Recently, the toxicity of C. serpens was described in ruminants and horses in the semi-arid region of Bahia and its photosensitizing action was confirmed in sheep [5]. In these initial studies the intoxication was produced in adult sheep in a pasture invaded by C. serpens. In this experiment only one necropsy was performed, and the animal showed liver lesions suggestive of hepatogenous photosensitization.” In lines 304-307 it may be found that “In the pioneering study [5], the probable hepatoxic action of C. serpens was based, equivocally, based on the increase in the activity of the GGT enzyme, as well as the observation of degenerative-necrotic alterations in liver biopsies and in two necropsied animals.” - How many necropsies were performed in this experiment? Where do these inaccuracies arise in the discussion of your own work?
Author Response
Dear editor and reviewers,
All changes were accepted and made as requested. we appreciate the suggestions, and these were scored individually to clarify the questions. The changes were highlighted in the text.
✓ University street name has changed, corrected in authors description
# Reviewer 02
The presented paper reports effect of experimental intoxication of sheep by ingestion of C. serpens in order to explain the type of photosensitization caused by this plant and to verify if it remains toxic after harvesting and storage. The manuscript seems to be clearly written, however experiment design is inappropriate or, at least, description of the methodology is insufficient and contains mistakes.
General remarks:
Lines 87-88: “Eight crossbred Santa Inês and Dorper sheep were used, one male (Ov1) and seven females (Ov2 to Ov8), with liveweight (LW)between 10 and 40 kg and age between with live weight (LW) between 10 and 40 kg and age between six and 36 months.” - The study was conducted on a very small number of sheep, in addition animals were differentiated in terms of age, sex, physiological status, coat color, etc. All these factors can influence light sensitivity. Itis well known that young animals and light-colored individuals are more susceptible to skin damage under the influence of sunlight. Ov2 that showed the most severe clinical symptoms was “newly weaned” and white. Paleasse argue that the experimental groups were representative.
✓ For approval of the experiment in the animal ethics committee, as photosensitization is a well-explained syndrome with other etiologies, it is recommended to use a reduced number of animals so that, by causing suffering to animals, few animals with repeatability of clinic signs even in different proportions are sufficient to clarify the pathophysiology. The objective was to clarify whether it was primary or secondary, as all animals, even if in different herald ranges and of both sexes, had skin and non-hepatic lesions, was sufficient to answer the hypothesis.
Lines 91-92: “During the day the sheep were kept in a sunnyarea without pasture and at night” - what does it mean? Partof the sentence is probably missing here. Additionally, please describe in more detail the conditions in which the animalswere kept. Were the animals from different groups kept inexactly the same conditions and in the same pen? Were theanimals able to hide in the shade? Were the animals trimmed prior to the start of the experiment? If so, are all of them? Atthe same time? In the same way?
✓ The text was corrected to clarify the methodology and how the animals were kept
Line 92: “The animals were dewormed before the beginning of the experiment” - Please provide detailed information ondeworming (name of the drug, dose, term of deworming),bacause antiparasitic agents can act photosensitizing.
✓ The treatment was inserted in the text, the drug chosen was chosen so as not to interfere with the methodology of the experiment
Line 101: “The plant was collected every 15 days, stored instrain bags, and kept in a cold chamber (3-5°C).” - Are theseconditions suitable for storing plants for animal nutrition?Were the feed rations for the control animals also stored inthis way? Is there a risk of microbial growth on plants duringcold storage, which could have an adverse effect on animalhealth?
✓ The methodology used was the same as that cited by Pimentel et al., 2007. Mendez et al., (1991) and Cruz et al., (2012), also kept the plants refrigerated, but they are not in our ref. are works with Ammi majus and Pterodon emarginatus.
Lines 117-118: “Four of the sheep were recently weaned andall the herd were maintained previous to the experiment in apasture free of Brachiaria spp and other hepatotoxic plants.” -Please indicate exactly which animals were just afterweaning. The body weight of 6 out of 8 individuals indicates that they were not individuals just after weaning. Why werethe youngest animals taken into group G1? Please explain it.
✓ Only two of the animals were newly weaned, data corrected in the text, and they were placed in group 1 precisely because of the ease of adapting them to the consumption of the legume and the volume of plant to be offered to be offered, different from adult animals.
Lines 130-133: “Serum samples were obtained after clotretraction, with centrifugation at 1200X g for 10 minutes and,later, placed in polyethylene microtubes (Ep-pendorf®) andsent for measurement of serum activities of aspartate-aminotransferase (AST) and gamma-glutamyl transferase (GGT)”. -Why did you not perform a biochemical analysis of totalbilirubin, direct bilirubin, indirect bilirubin, while your pilotstudy (referred to in reference 5) showed significant changesin these parameters in sheep experimentally intoxicated with C. serpens? Please explain it.
✓ They were not performed because in the experiments after the pilot in which there was repeatability of sample collection, these parameters did not change, for this reason they were not performed in this experiment.
Lines 144-148: "During the necropsy, fragments of the skinfrom different regions (with and without lesions), organs fromthe abdominal, thoracic, and pelvic cavities, and the brainwere collected. This material was fixed in 10% formalin,processed by routine histological techniques, stained withhematoxylin and eosin (HE) and evaluated under an opticalmicroscope.” - What was the purpose of taking samples ofthe brain and organs from the abdominal, thoracic, and pelviccavities? What kind of changes were expected to be foundthere? Please explain it.
✓ It is a protocol of the laboratory of veterinary pathology of the Federal University of Bahia, to evaluate all organs when necropsies were performed so as not to have direction for the lesions that are observed only macroscopically.
Line 231: Please provide more information on reference standards for the parameters described; do they take intoaccount age, gender, etc? What is more, I have not found inthe quoted paper such a range of standards for sheep inrelation to GGT...
✓ The experiment was carried out in Brazil, country of continental proportions in which there are variations between the different climate biomes. In this way, there are comparisons with international parameters such as from Kaneko's book, which are often not appropriate, so in addition to these parameters from the literature, we used the animal itself at the beginning of the experiment, under healthy conditions the basal control of biochemical values.
✓ The Kaneko book reference has been added to the reference to clarify the origin of the data.
Line 287-288: “including newly weaned sheep (without previous consumption of any plants or potentially hepatotoxic substances)” - Is it possible for lambs around 180 days (6months) old to eat exclusively mother's milk? Have the “newly weaned lambs” been in any contact with their mothers? Is it possible that the changes observed in behavior of sheep from G1 group were related to the weaning effect? Please explain it.
✓ The expression is that up to six months these animals were not kept exclusively on breast milk, but rather that these animals had a controlled diet so that they did not have contact with any plant that was potentially toxic and that could alter the observations of the experiment. they ate like the mother elephant grass (Pennisetum purpureum). as there was a period of adaptation of the animals before the experiment, including the lambs, it is not possible to associate the changes in behavior with separation from the mother.
Minor remarks:
1. Lines 31-32: “During 30 days, G1 received an exclusivediet of C. serpens, and G2 and G3 received 31 10g/kg/dand 20g/kg/d, respectively” - It should be mentioned thatC. serpens was added on kg of body weight; it is notclear in this statement.
✓ Corrected as suggested
2. Line 33: “for x-xx days” - what does it mean?
✓ Corrected item in text
3. In lines 88-89 is written “sheep […] with live weight (LW)between 10 and 40 kg”, while in Table 1 where is noticedthat the lowest weight of sheep was 13 kg (final weightof Ov2) and the highest weight was 50 kg (final weightof Ov7). Mature weights of the Santa Ines ewes fallbetween 40 and 50 kg. The Dorper is the second largestbreed in South Africa and adult ewes usually weigh 60-80 kg. What is typical weight of adult crossbred SantaInês x Dorper sheep? Please provide reliableinformation about the weight of sheep.
✓ We consulted the professor responsible for the herd of the university from which the animals came, so that the information could be corrected in the appropriate way, the animals in the herd are the Santa Ines breed of the common type in which adult sheep have an average weight between 40 and 50 kg.
✓ The information about the correct weight and breed of the animals to be compatible has been corrected in the text.
4. Lines 149-150: „Additionally, after cessation of plantadministration, Ov1 from G1 and Ov7 from GC wererelocated to a shaded paddock and followed for another30 days to assess the recovery of skin lesions of Ov7.” -ArethereanyskinlesionsinOv7?Why?
Are there any skin lesions in Ov7? Why?
✓ Error in typing the text, sheep 1 that recovered from the lesions and sheep 7 was a control animal. information has been corrected in the text.
5. Lines 280-282: “However, in this same study, [14] theauthors drew attention to some inconsistencies: the appearance of clinical signs was very fast, lethality innatural cases was rare and there was rapid recoveryafter removal of the animals from the pastures invadedby the plant.” - It is not the same study as cited inprevious lines. It focused on other plant Malachrafasciata.
✓ Corrected item in text
In lines 276-280 is written “Recently, the toxicity of C.serpens was described in ruminants and horses in thesemi-arid region of Bahia and its photosensitizing actionwas confirmed in sheep [5]. In these initial studies theintoxication was produced in adult sheep in a pastureinvaded by C. serpens. In this experiment only one necropsy was performed, and the animal showed liverlesions suggestive of hepatogenous photosensitization.”In lines 304-307 it may be found that “In the pioneeringstudy [5], the probable hepatoxic action of C. serpenswas based, equivocally, based on the increase in theactivity of the GGT enzyme, as well as the observationof degenerative-necrotic alterations in liver biopsies andin two necropsied animals.” - How many necropsieswere performed in this experiment? Where do these inaccuracies arise in the discussion of your own work?
✓ in the previous experiment only one necropsy was performed. this information has already been corrected in the discussion text.
Dear editors and reviewers, we appreciate all grammatical corrections and suggestions for adjustments, all requests have been made and we hope that they meet the needs and reiterate the estimate that the article will be added to the scope of the journal. Additionally the references were thoroughly reviewed.
Best regards,
Tiago da Cunha Peixoto and team.

Round 2
Reviewer 2 Report
After reviewing the Authors' answers and corrections in the manuscript, I have no more questions or comments.
Author Response
Dear reviewer,
We appreciate all the considerations and look forward to considering the article for publication in the journal.
As there were no further requests for adjustments, we thank you for your cooperation and remain at your disposal for the editors.